# Osteogenic Commitment of Human Periodontal Ligament Cells Is Predetermined by Methylation, Chromatin Accessibility and Expression of Key Transcription Factors

**DOI:** 10.3390/cells11071126

**Published:** 2022-03-26

**Authors:** Rahyza I. F. Assis, Francesca Racca, Rogério S. Ferreira, Karina G. S. Ruiz, Rodrigo A. da Silva, Samuel J. H. Clokie, Malgorzata Wiench, Denise C. Andia

**Affiliations:** 1Department of Prosthodontics and Periodontics, Piracicaba Dental School, University of Campinas, Piracicaba, São Paulo 13414-018, Brazil; r162379@dac.unicamp.br (R.I.F.A.); f146101@dac.unicamp.br (F.R.); kgsilverio@fop.unicamp.br (K.G.S.R.); 2School of Dentistry, Institute of Clinical Sciences, Institute of Cancer and Genomic Sciences, University of Birmingham, Birmingham B5 7EG, UK; 3School of Dentistry, Health Science Institute, Paulista University, São Paulo 04026-002, Brazil; rogerio.ferreira25@aluno.unip.br; 4Program in Environmental and Experimental Pathology, Paulista University, São Paulo 04026-002, Brazil; rodrigo.silva3@docente.unip.br; 5West Midlands Regional Genetics Laboratory, Birmingham Women’s and Children’s Hospital, Birmingham B15 2TG, UK; s.clokie@nhs.net

**Keywords:** periodontal ligament cells, osteogenesis, DNA methylation, transcriptome, epigenomics

## Abstract

Periodontal ligament stem cells (PDLCs) can be used as a valuable source in cell therapies to regenerate bone tissue. However, the potential therapeutic outcomes are unpredictable due to PDLCs’ heterogeneity regarding the capacity for osteoblast differentiation and mineral nodules production. Here, we identify epigenetic (DNA (hydroxy)methylation), chromatin (ATAC-seq) and transcriptional (RNA-seq) differences between PDLCs presenting with low (l) and high (h) osteogenic potential. The primary cell populations were investigated at basal state (cultured in DMEM) and after 10 days of osteogenic stimulation (OM). At a basal state, the expression of transcription factors (TFs) and the presence of gene regulatory regions related to osteogenesis were detected in h-PDLCs in contrast to neuronal differentiation prevalent in l-PDLCs. These differences were also observed under stimulated conditions, with genes and biological processes associated with osteoblast phenotype activated more in h-PDLCs. Importantly, even after the induction, l-PDLCs showed hypermethylation and low expression of genes related to bone development. Furthermore, the analysis of TFs motifs combined with TFs expression suggested the relevance of SP1, SP7 and DLX4 regulation in h-PDLCs, while motifs for SIX and OLIG2 TFs were uniquely enriched in l-PDLCs. Additional analysis including a second l-PDLC population indicated that the high expression of *OCT4*, *SIX3* and *PPARG* TFs could be predictive of low osteogenic commitment. In summary, several biological processes related to osteoblast commitment were activated in h-PDLCs from the onset, while l-PDLCs showed delay in the activation of the osteoblastic program, restricted by the persistent methylation of gene related to bone development. These processes are pre-determined by distinguishable epigenetic and transcriptional patterns, the recognition of which could help in selection of PDLCs with pre-osteoblastic phenotype.

## 1. Introduction

Stem cell-based treatments have become increasingly promising in the regeneration of tissue lost due to a disease. Adults have several sources of stem cells, such as bone marrow (BMSCs), adipose tissue (ASCs) and teeth (periodontal ligament cells-PDLCs, dental pulp cells-DPCs and stem cell from exfoliated human dentition-SHED). Previous studies have shown that PDLCs have mesenchymal stem cells properties, such as self-renewing; ability to differentiate into osteoblast, adipocyte and chondrocyte-like cells; and the expression of specific stem cell surface markers [1,2,3,4]. Moreover, the acquisition of PDLCs is easier than other sources, especially when compared to BMSCs [4]. Although studies have shown similarities between PDLCs and BMSCs related to their stem cells properties, PDLCs might present higher heterogeneity in producing mineral nodules when induced for osteogenic differentiation in vitro [5,6]. The mechanisms underlying this heterogeneity remain poorly understood.

Previous studies have also demonstrated that the osteogenic differentiation potential is influenced by cells’ epigenetic landscape [7,8]. Epigenetic regulation is responsible for adding a new layer into a complex regulatory network during stem cell differentiation by regulating gene expression through chromatin accessibility or non-coding RNAs [9]. One of the most important epigenetic mechanism is DNA methylation [10], which is responsible for adding a methyl residue at carbon 5 of cytosine in a CpG dinucleotide-context [11,12]. DNA methylation plays key role in the regulation of osteogenesis [13] while PDLCs express higher osteogenic potential and an enhanced mineralization capacity in vivo compared with others dental cells [10]. When DNA methylation occurs in gene promoter regions, it correlates with gene silencing [14]. Although its role in other gene regions is less clear, methylation also correlates with decreased chromatin accessibility and transcription factors (TF) binding at gene regulatory regions, including distant enhancers [15,16,17]. In addition, DNA hydroxymethylation is a modification indicative of active DNA demethylation processes [18]. Chromatin accessibility is the most commonly assessed genome-wide through ATAC-seq (Assay for Transposase-Accessible using sequencing), which also allows for identification of enriched TF motifs within the selected regions. Such chromatin structure-led TF activity establishes cell-type specific gene expression [19,20,21] and, therefore, cell type-specific transcriptional programs and cell commitment [22]. Importantly, epigenetic modifications are reversible either by certain cell stimuli or epigenetic drugs, resulting in changes in gene transcription [23].

This study aims to identify transcriptional programs related to osteogenic potential in PDLCs and to establish whether they are dictated by DNA (hydroxy)methylation and chromatin accessibility. Our findings indicate distinct maps of DNA methylation and chromatin accessibility between l-PDLCs (low osteogenic potential) and h-PDLCs (high osteogenic potential) associated with differential transcriptional regulation and gene expression patterns.

## 2. Materials and Methods

### 2.1. Cell Acquisition and Characterization

PDLCs were harvested from extracted third molars from two 20–22-year-old subjects after signing an informed consent approved by the Ethics Committee of Piracicaba Dental School, University of Campinas, São Paulo, Brazil (CAAE55588816.4.0000.5418). Periodontal ligament tissue isolation and cell culture were performed as previously described [24]. PDLCs were characterized according to Dominici et al. [25] to confirm the ability to differentiate into osteogenic and adipogenic cell lineages and the expression/lack of expression of specific cell surface markers, such as CD166, CD34 and CD45 [26]. The levels of CD34 and CD45 were very similar between h- and l-PDLCs, showing less than 1% of expression of positive cells. Regarding multipotency marker CD166, more than 95% of cells in both populations showed positive expression (data not shown). Alizarin red staining was performed to assess the amount of mineral matrix produced in vitro by each cell population [5]. Consequently, PDLCs were classified either as high osteogenic potential PDLCs (h-PDLCs), which was the cell population with capacity to produce higher amounts of mineral matrix, or low osteogenic potential PDLCs (l-PDLCs, with lower capacity to produce mineral matrix) [5,8]. All experiments were performed between passages 5 and 6. Based on our previous studies [5,7], we chose day 10 of the osteogenic media (OM) induction as the time point to analyze epigenomic and transcriptomic changes (Figure 1).

### 2.2. Cell Culture and Osteogenic Stimulation

In order to investigate epigenetic changes at basal levels and upon osteogenic differentiation, two PDLCs classified as l-PDLCs and h-PDLCs according to the above description were plated at 8.7 × 10^5^ cells per 100 mm dishes either in Dulbecco’s Modified Eagle Medium (DMEM), containing 10% fetal bovine serum (FBS), penicillin (100 U/mL) and streptomycin (100 mg/mL) (Gibco, Carlsbad, CA, USA) (DMEM group) or in osteogenic medium (OM) (Lonza, Walkersville, MD, USA) supplemented as above (OM group), with media change every 3 days.

### 2.3. mRNA Isolation and RNA-Seq

l-PDLCs and h-PDLCs were cultured at 1.5 × 10^5^ cells per well in 6-well plate in DMEM or OM, changing the media every 3 days. After 10 days, the culture medium was removed, and cells were washed with PBS and scrapped off in TRizol reagent (Gibco BRL, Rockville, MD, USA). Total RNA was purified following the manufacturer’s instructions and RNA samples were stored at −80 °C. RNA concentration and quality were assessed using Qubit (Thermo Fisher Scientific Inc., Rockford, IL, USA) and a spectrophotometer (Nanodrop 1000; Nanodrop Technologies LLC, Wilmington, NC, USA). The samples for each experimental condition were initially obtained from three independent experiments and then pooled in equal concentrations and sequenced using Illumina TruSeq Stranded mRNA Sample Prep Kit in Illumina NextSeq 500 platform (Illumina Inc., San Diego, CA, USA) in the Genomics Birmingham Facility (Birmingham, UK).

### 2.4. RNA-Seq Data Processing

The reads obtained after sequencing were aligned to the human genome (hg19) using HiSAT2 and processed with bedtools to generate normalised coverage plots. Quantification was performed according to the latest recommended pipeline as defined in the DeSeq2 software. A count for each gene was calculated using the reference-free aligner Salmon [27] and the resulting count table was processed using DeSeq2 [28] to compare the treatment groups. Genes with log2 fold change above 1.5 or below −1.5 were considered differentially expressed. The log2 fold change was used to generate heatmap charts. Each group (DMEM and OM) was subjected to gene ontology (GO) analysis using the Gene Ontology Consortium software [29] followed by the removal of redundant terms using REVIGO [30]. The top 5 terms with the Log10 *p*-value ≤ 3.0 from each group were collated. GO-REVIGO analysis was performed on gene sets that were upregualted and downregulated in l-PDLCs and h-PDLCs, and the top terms were ranked by Log10 *p*-values and shown as bar graphs. Exploratory data analysis was performed on RNA-Seq data using principal component analysis (PCA). The plot PCA function from the DESeq2 package was used to perform the PCA and to plot the top two principal components.

### 2.5. Real Time q-PCR RNA Analysis

*OCT4*, *SIX3*, *PPARG* and *SP7* genes were selected to test their predictive osteogenic value using an independent population of l-PDLCs (l-PDLCs-2, see Assis et al. [5], Appendix A). h-PDLCs, l-PDLCs and l-PDLCs-2 were cultured for RNA extraction, as described above and cDNA synthesis was performed using 1 μg RNA, as described previously [27]. Quantitative PCR was carried out using LightCycler 480 Real-Time PCR System (Roche Diagnostics GmbH, Mannheim, Germany), FastStart Essential DNA Green Master kit (Roche Diagnostic Co., Indianapolis, IN) and primers indicated in the Appendix A, according to the manufacturer’s instructions and in technical triplicates. The results of three biological replicates were analyzed by the ∆∆Ct method [31] and are presented as gene expressions relative to *ACTB* as a reference gene.

### 2.6. Assay for Transposase-Accessible Chromatin Using Sequencing (ATAC-Seq)

A total of 5 × 10^4^ cells were harvested from l-PDLCs and h-PDLCs under control conditions (DMEM group) or osteogenic induction (OM group) and were incubated in transposition reaction as preconized by Buenrostro et al. [32]. The Tn5 enzyme recognizes regions with open chromatin and cut these regions. Digitonin was included to reduce contamination with mitochondrial DNA [33]. The cell pellet was gently pipetted to resuspend the transposition mix and incubated at 37 °C for 30 min; then, the samples were purified using MinElute PCR Purification Kit (Qiagen, UK) and amplified as described in previous study [32]. Adapters containing unique index sequences were added to allow the libraries from different samples to be pooled and individually identified during downstream analysis. The samples were sequenced using Illumina NextSeq 500 platform (Illumina Inc., San Diego, CA, USA) in the Genomics Birmingham Facility (Birmingham, UK). ATAC-seq data were obtained from two independent experiments.

### 2.7. ATAC-Seq Genome Alignment and Peak Calling

Using HOMER software [34], low quality reads, i.e., any read with a Phred quality score of less than 30 which points to ambiguous nucleotide callings, were removed. The Bowtie2 tool was used to align the reads to the hg19 version of the human genome. Any duplicated and ENCODE blacklisted reads were also removed. Non-uniquely mapped reads were filtered out and the peaks were called using the ‘factor mode’ in HOMER based on default settings. The bedtools suite was used to calculate coverage maps, for which its output includes reads mapped to chromosome number and the coverage depth (the number of reads for each nucleotide). Bigwig files were also generated to enable data viewing using the UCSC Genome Browser.

### 2.8. TF Motif Analysis 

Enrichment for potential TF binding sites within ATAC-seq peaks was identified by using motif analysis using the ‘findMotifsGenome’ script within HOMER. Enriched motifs were identified by calculating the frequency of target sequences versus background sequences (50,000 randomly selected genomic sequences). TF motifs were identified based on ‘homermotifs’ setting. Firstly, motifs with a score above 0.8 were sorted by *p*-value and the top 17 motifs were retained and sorted by percentage of target (% of target), which indicates the prevalence of a motif in the identified ATAC-seq peaks.

### 2.9. DNA Isolation 

After 10 days of osteogenic induction, the culture medium was removed, the cells were washed two times with PBS and total DNA was purified by extraction with phenol/chloroform/isoamyl alcohol (25:24:1 *v/v*, Thermo Fisher Scientific Inc. Rockford, IL, USA) and DNA samples were stored at −20 °C. DNA concentration and quality were assessed using Qubit (Thermo Fisher Scientific Inc., Rockford, IL, USA) and spectrophotometer (Nanodrop 1000; Nanodrop Technologies LLC, Wilmington, NC, USA). Three independent experiments were performed, and these samples were employed in the DNA (hydroxy)methylome epigenetic analyses as described below.

### 2.10. Oxidative Bisulfite Conversion and DNA (Hydroxy)Methylome 

The oxidative bisulfite conversion reaction was performed using TrueMethyl oxBS Module (catalog #0414, NuGEN, Tecan Genomics, Inc., Redwood City, CA, USA). Samples from the control DMEM groups (DMEM) were pooled separately for both l-PDLCs and h-PDLCs, combining 500 ng of each replicate. For the induced group (OM), the replicates were run independently, totaling 3 OM samples for each PDLC population. Then, 1 µg of DNA was purified and denatured, according to the manufacturer’s specifications. DNA from each group was split in two equal tubes of reactions, one of which underwent chemical oxidation followed by bisulfite conversion, the other underwent mock oxidation (oxidant replaced by water) followed by bisulfite conversion. This allows distinguishing between DNA methylation and hydroxymethylation. Next, the Infinium Methylation EPIC BeadChip (Illumina Inc., San Diego, CA, USA) kit was employed, and all reactions were processed according to EPIC array protocol. Array bead chips were scanned on Illumina HiScan SQ System (Illumina Inc., San Diego, CA, USA).

### 2.11. (Hydroxy)Methylation Data Processing

The data were processed using R statistical environment through minfi package [35,36], associated with dplyr and tidyr packages [37]. Data normalizations were performed using quartiles methods. Probes were considered differentially methylated when delta beta > 0.2 (hypermethylated) or delta beta < −0.2 (hypomethylated) and *p*-value < 0.01. Delta beta values represent the measured (hydroxy)methylation values, based on the intensities of probes. Graphs were built using ggplot2 [38], VennDiagram [39]. The generated gene list was subjected to gene ontology analysis using Database for Annotation, Visualization, and Integrated Discovery (DAVID) followed by REVIGO (Reduce and Visualize Gene Ontology) to remove redundant terms [30].

## 3. Results

### 3.1. l-PDLCs Are Characterised by Neuronal Rather Than Osteoblastic Cell Pre-Commitment

The osteogenic potential of h-PDLCs and l-PDLCs was previously established based on the cells’ ability to produce mineral modules [5]. However, we hypothesize that this potential is determined before PDLCs are stimulated toward osteogenesis. To characterize transcriptional activity occurring at basal levels, PDLCs presenting distinct osteogenic potentials were subjected to RNA-seq and differential expression analysis (Figure 2). Differentially expressed genes included 1396 (8.45%) genes upregulated in l-PDLCs and 984 (5.96%) genes upregulated in h-PDLCs (Figure 2A), as determined by log2FoldChange (>1.5). The top 60 most differentially expressed genes are clearly distinguish between the two populations (Figure 2B and Appendix A) and include transcripts for several TFs known to be involved in cell fate commitment. *SP7 transcription factor* (*SP7*) and *Distal-Less Homeobox 4* (*DLX4*) are related to the positive regulation of osteoblast differentiation; here, they indeed show significantly higher expression in h-PDLCs. In comparison, *Iroquois Homeobox 6* (*IRX6*) and *SIX Homeobox 3* (*SIX3*), genes that play a role in proliferation and differentiation of neuronal progenitor cells, were found to be upregulated in l-PDLCs. Biological processes related to the differentially expressed genes were also identified. The transcriptional pattern in l-PDLCs indicates cell adhesion, cell surface receptor linked signalling and G-protein signalling (Figure 2C). On the other hand, organ development, immune response and cell–cell signalling are the most activated pathways in h-PDLCs (Figure 2D). Although the multicellular organismal process has been activated in both PDLCs, only h-PDLCs showed more biological processes associated with this cascade, such as organ development.

These results show that l-PDLCs and h-PDLCs have distinct transcriptional regulation at basal levels. l-PDLCs are characterized by the upregulation of metabolic processes, tissue homeostasis and genes associated to other, especially neuronal, cell lineages, while the h-PDLCs’ transcriptome is enriched in system and organ development and cell communication, with the key transcriptional regulators related to osteoblast differentiation being upregulated.

### 3.2. Osteogenic Commitment in PDLCs Is Predefined by Chromatin Accessibility and DNA Methylation

Chromatin accessibility and DNA (hydroxy)methylation levels were next studied to characterise epigenetic states supporting the different basal transcription profiles in PDLCs presenting distinct osteogenic potential. ATAC-seq analysis resulted in identification of 102,122 accessible chromatin regions in l-PDLCs and 25,896 in h-PDLCs (Appendix A). Although the ATAC-seq regions had similar peak scores and distributions in relation to the transcriptional start sites (TSS) in both l-PDLCs and h-PDLCs (Appendix A), h-PDLCs were characterised by a higher percentage of peaks located within promoters (Appendix A). Due to this and the difference in the number of peaks identified in l- and h-PDLCS both before and after osteogenic stimulation (Appendix A), the two populations were not compared directly through differential analysis. Instead, the pathway enrichment analysis was performed using the top 10% peaks in each experimental group. This analysis shows that both PDLCs share common biological process, such as cell cycle (GO:0007049) and cell division (GO:0051301) (Figure 3A,B). However, open chromatin regions in l-PDLCs are associated with genes involved in histone deacetylation (GO:0016575) and negative regulation of TOR signalling (GO:0032007) (Figure 3A), while in h-PDLCs, the peaks were associated with small GTPase mediated signal transduction (GO:0007264) and ERAD pathway (GO:0036503), both related to osteoblast activation and differentiation (Figure 3B) [40,41,42].

DNA methylation is essential for the regulation of tissue-specific genes, while DNA hydroxymethylation points towards dynamic changes in local methylation levels. Here, both DNA methylation and hydroxymethylation analyses were performed using the Illumina Infinium Methylation EPIC BeadChip assay involving 850,000 CpG sites (Figure 3C, Appendix A). The DNA modification levels were compared between l-PDLCs and h-PDLCs, and the analyses indicated overall similar DNA methylation patterns at basal levels in both populations, while overall hydroxymethylation levels appeared slightly higher in h-PDLCs (Figure 3C, Appendix A). Only 17,847 (2.1%) probes showed differential DNA methylation, with more hypomethylated probes 10,276 (57.6%) compared to 7571 (42.4%) hypermethylated probes in l-PDLCs (Figure 3D). The genes linked to probes hypermethylated in l-PDLCs were subjected to pathways enrichment analysis and showed enrichment for biological process such as positive regulation of bone mineralization (GO:0030501) (Figure 3E). The genes associated with this process include *Myocyte Enhancer Factor 2C (MEF2C)*, *CD276 Molecule* (*CD276)*, *Fibrillin 2* (*FBN2)*, *Solute Carrier Family 8 Member A1* (*SLC8A1)* and *Odd-Skipped Related Transcription Factor 1* (*OSR1)*. In agreement with increased DNA methylation, most of the transcripts of these genes were downregulated in l-PDLCs in RNA-seq data (Figure 3F). Furthermore, in agreement with RNA-seq data (Figure 2), the probes hypomethylated in l-PDLCs were associated with genes involved in neuronal and synaptic processes (Appendix A).

All three methodologies point towards less active osteogenic processes and genes in l-PDLCs when compared to h-PDLCs (Figure 3G,H). A total of 491 genes were identified through an overlap of upregulated genes, genes linked to accessible chromatin and genes linked to hypomethylated probes in l-PDLCs (Figure 3G). These genes represent the processes active in l-PDLCs and linked to signal transduction (GO:0007165), synapse organization (GO:0050808) and excitatory postsynaptic pathways (GO:2000463), again indicating neuronal commitment (Figure 3H).

In summary, these results show that l-PDLCs and h-PDLCs have distinct epigenetic regulation at basal levels, which could contribute to their differentiation commitment. Whereas h-PDLCs showed open chromatin in gene regions associated with osteoblast activation and differentiation, l-PDLCs showed hypermethylation of genes related to the positive regulation of bone mineralization biological process. The results obtained so far suggest presence of epigenetic memory that interferes with the transcriptional activation of osteogenic genes in l-PDLCs, contributing to the delay in osteogenic differentiation.

### 3.3. Genes and Biological Processes Related to Acquisition of Osteoblast Phenotype Are Further Activated in h-PDLCs upon Osteogenic Stimulation

The effect of osteogenic induction on gene expression and biological processes was subsequently investigated in both PDLCs. After 10 days of osteogenic stimulation (OM), a total of 963 genes were upregulated and 1223 were downregulated in l-PDLCs, whereas 1103 genes were upregulated and 1547 were downregulated in h-PDLCs (log2 fold change > 1.5) (Figure 4A,B). The upregulated genes were subjected to network analysis to identify biological processes triggered by them (Figure 4C,D). Following osteogenic induction in l-PDLCs, the network remained broad but, interestingly, highlighted the activation of multicellular organismal development (GO:2000026) and immune response (Figure 4C), previously observed in h-PDLCs at basal levels (Figure 2D). This was also accompanied by the activation of cell communication (GO:0007154) and cell–cell signalling (GO:0007267) (Figure 4C). In contrast, in h-PDLCs, osteogenic stimulation activated biological processes such as cell cycle (GO:0007049) and DNA replication (GO:0006260) (Figure 4D). Several genes important for osteoblastic phenotype acquisition were only upregulated in h-PDLCs and these include *Bone Morphogenetic Protein 7* (*BMP7*) and *Claudin 14* (*CLDN14*) (Figure 3E,F). Although osteogenic induction clearly separated the two populations according to their transcription profiles (PCA analysis, Appendix A), the expression pattern of the top 100 most upregulated genes (Figure 4E and Appendix A) was not as striking as observed at basal levels (Figure 2B). Furthermore, osteogenic induction resulted in a common upregulation of 197 genes, including genes involved in the ossification process (Figure 4G). However, the transcriptional data related to this specific biological process revealed that the upregulation of some key genes related to osteogenesis such as *Sclerostin* (*SOST*), *Forkhead Box C2* (*FOXC2*) and *ATPase H+ Transporting V0 Subunit A4* (*ATP6V0A4*) was higher in h-PDLCs than in l-PDLCs (Figure 4H). Additionally, the levels of osteogenic markers previously identified by Javed et al. [43]) also show a general pattern of upregulation in h-PDLCs in comparison to l-PDLCs (Appendix A).

Altogether, osteogenic stimulation triggers biological processes related to osteoblast phenotype acquisition in both populations; however, they remain less active in l-PDLCs. Moreover, 10 days of osteogenic stimulation led the l-PDLCs to a similar transcriptional profile. as observed in h-PDLCs at basal levels, whereas h-PDLCs potentially proceeded to the next stage of osteoblast differentiation.

### 3.4. Transcriptional Changes upon Osteogenic Induction Are Not Accompanied by Major Epigenetic Restructure

After 10 days of osteogenic stimulation, less than 1% of probes became differentially methylated in both PDLCs (Figure 5A,B). Interestingly, these DMPs showed different patterns of DNA methylation changes among l- and h-PDLCs, i.e., about 90.25% of DMPs were hypomethylated in h-PDLCs whereas only 66.3% were hypomethylated in l-PDLCs (Figure 5B) with the majority of DMPs located at gene bodies and, therefore, outside the promoters (Figure 5C). To further investigate these differences, we inspected the l-PDLCs-specific DMPs in h-PDLC data by comparing the delta β values. The majority of DMPs in l-PDLCs appear to be invariant following 10 days of osteogenic differentiation in h-PDLCs (Delta β = 0) (Figure 5D). The same analysis was performed for h-PDLCs-specific DMPs; here, these hypomethylated DMPs presented similar patterns in l-PDLCs (Figure 5E). The overall levels of hydroxymethylation did not change in h-PDLCs but slightly increased in l-PDLCs (Appendix A).

These data suggest that DNA methylation changes that follow osteogenic stimulation are limited in both PDLC populations, although they appear more dynamic in l-PDLCs. Therefore, the next step was to investigate if these involve regions associated with key osteogenic genes.

### 3.5. Progression in Osteoblast Phenotype Following Osteogenic Induction Is Limited by Persistent DNA Hypermethylation in l-PDLCs

The most relevant differences between h-PDLCs and l-PDLCs were observed when DNA methylation and chromatin accessibility were compared between the two OM data sets (Figure 6). Probes numbering 14,194 (1.66%) were differentially methylated with 8740 of them (61.6%) hypomethylated and 5454 (38.4%) hypermethylated probes in l-PDLCs compared to h-PDLCs (Figure 6A). Pathway enrichment analysis showed that genes involved in bone development (GO:0060348) remained hypermethylated in l-PDLC even upon stimulated conditions (Figure 6B). In agreement with DNA methylation levels, the genes associated with this biological process showed higher expression in stimulated h-PDLCs than l-PDLCs (Figure 6C).

### 3.6. Chromatin Accessibility Supports Gene Transcription Related to Osteoblast Function in Stimulated h-PDLCs

Open chromatin regions detected after stimulation also confirmed the different levels of commitment to osteogenic pathways in the two PDLCs populations. In l-PDLCs, they were associated with biological processes related to WNT signalling pathways (GO:0016055), DNA replication (GO:0006260) and cell division (GO:0051301) (Figure 6D), whereas the activations of the response to unfolded proteins (GO:0006986), ERAD pathway (GO:0036503), transforming growth factor beta (GO:0007179) and osteoblast differentiation (GO:0001649), all of which are associated with progression to osteoblast phenotype, were enriched in h-PDLCs (Figure 6E). This is reflected by upregulated gene expressions in h-PDLCs for the gene sets related to the response to unfolded proteins (Figure 6F) and to osteoblast differentiation (Figure 6G).

Taken together, these data suggest a faster progression of h-PDLCs towards osteoblast commitment supported by the initial establishment of chromatin and epigenetic patterns.

### 3.7. Non-Osteogenic TFs Are Uniquely Involved in l-PDLCs and Could Be Used to Predict Low Osteogenic Potential

The RNA-seq analysis at basal levels pointed towards the involvement of TFs implicated with cell commitment programmes (Figure 2B). Therefore, we interrogated the ATAC-seq data sets for motif enrichment to identify potential TFs involved in each of the PDLCs population (Figure 7A). TF motifs are short DNA sequences specifically recognised and bound to by transcription factors [44]. The analysis revealed an enrichment for a common core of TFs: SP1/GC-box, RUNX, AP1 and TEAD (Figure 7), which agrees with Tarkkonen et al. [45]. However, h-PDLCs showed higher dependence on SP1 at both basal and induced levels in comparison to l-PDLCs. Importantly, the analysis also pointed towards motifs uniquely observed for l-PDLCs (SIX, OLIG2) at basal levels (Figure 7A).

The expression levels of potential TFs associated with the motifs were also collected from the RNA-seq data at basal and stimulated conditions (Figure 7B). Several TFs such as *DLX4* and *SP7* appear to be highly expressed in h-PDLCs, *OLIG2* and *SIX1* at DMEM in l-PDLCs and OM in h-PDLCs and *KLF4* at OM in h-PDLCs.

The data presented so far suggest that l-PDLCs are characterised by delayed and incomplete osteogenic commitment, at least partially associated with persistent pluripotency and neuronal gene expression. Indeed, the high expression levels of *OCT4* and *SIX3* TFs were confirmed in the second population with low osteogenic potential (l-PDLCs-2), while the first l-PDLC population was characterised by high expression levels of adipogenic marker *PPARG* (Figure 7C). In agreement with RNA-seq data, the expression of osteogenic-specific *SP7* TF was higher in h-PDLCs than in the two l-PDLCs, although this difference was not statistically significant for the second l-PDLC population (Figure 7C). These observations suggest that high *OCT4*, *SIX3* or *PPARG* expressions might be better predictors of poor osteogenic commitment.

These initial results are important for better understanding of transcriptional control of the distinct osteogenic commitment in both l- and h-PDLCs.

## 4. Discussion

PDLCs are a valuable and accessible source of cells to be used in regenerative therapies. However, little is known about the impact of the epigenetic regulation on their gene transcription and, ultimately, the osteoblastic phenotype acquisition. Here, we provide a comprehensive genome-wide analysis of PDLCs with distinct osteogenic phenotypes, in an attempt to provide insights on how epigenetic regulation might affect the acquisition of osteoblast phenotype.

Firstly, at basal levels, both l- and h-PDLCs presented with distinct patterns of gene regulation, confirmed by transcriptome and ATAC-seq data, which suggested potential differential lineage commitment at the onset of osteogenic differentiation by progenitors in l-PDLCs. While several genes and biological processes related to G-protein receptor, cell adhesion and neuronal lineage were activated in l-PDLCs, h-PDLCs already showed activation related to osteoblast lineage and the progression to mature phenotype. The chromatin accessibility map also shows distinct pre-commitments between PDLCs’ populations. The mTOR osteogenic signalling pathway [46] was negatively regulated in l-PDLCs alongside the enrichment for histone deacetylation pathway, known by its repressive role during gene expression of *RUNX2* [47,48]. At the same time, h-PDLCs showed open chromatin regions associated with small GTPase-mediated signal, responsible for activating the Rho family that has been implicated as an alternative to Wnt signal transduction in osteogenesis [49]. Since osteoblast cells produce many extracellular matrix proteins, the activation of the ERAD pathway in h-PDLCs is essential to finely tune the control of protein conformation ahead of secretion [41]. The differences in the number of accessible chromatin region consistently observed between the l- and h-PDLCs could also be indicative of more differentiated and restricted regulation in h-PDLCs.

In addition, although both PDLCs share similar patterns of DNA methylation, the genes involved in the crucial biological process “bone mineralization (GO:0030282)” are hypermethylated in l-PDLCs at basal conditions, unlike in h-PDLCs. In summary, the osteoblast phenotype acquisition involves the induction of many genes that are already potentially active in mesenchymal stem cells [50]; here, this scenario is only observed at basal levels in h-PDLCs. In l-PDLCs, the lineage commitment is potentially skewed towards neuronal as indicated by the gene overlap of the three methodologies. Interesting, the presence of stem cell niche with neuronal commitment within PDL has been previously suggested [51].

Secondly, the analysis also revealed crucial differences between PDLCs under osteogenic induction. Specifically at day 10 of osteogenic stimulus, biological pathways enriched in l-PDLCs became similar to those observed in h-PDLCs at basal levels, highlighting the biological process activation of “multicellular organismal (GO:0051240)”; simultaneously, in h-PDLCs, “DNA replication (GO:0006260)” and “cell cycle (GO:0007049)” processes are most highly enriched, indicating the cells are at a highly proliferative stage of cellular differentiation [43]. Although both PDLCs shared the upregulation of genes related to “ossification (GO:0001503)”, their expression was much higher in h-PDLCs. Additionally, l-PDLCs showed DNA hypermethylation specifically in genes related to bone development, which might further explain the impairment of l-PDLCs to produce mineral nodules in vitro. This was also associated with their lower expression in l-PDLCs, confirming that DNA hypermethylation in those regions could affect transcript levels. Indeed, we have previously shown that the DNMT1 inhibitor, RG108, can partially restore *RUNX2* expression and enhance mineralisation in l-PDLCs [5]. The accessibility map in l-PDLCs at induced levels showed the activation of some biological processes previously observed in h-PDLCs at basal levels, suggesting that the initial epigenetic commitment to other lineages in l-PDLCs is partially overcome by osteogenic stimulation. In stimulated h-PDLCs, genes characterised by open chromatin region are associated with “osteoblast differentiation (GO:0001649)” and “response to unfolded protein (GO:0006986)”, a biological process essential for osteoblasts that continuously copes with the burden of protein synthesis overload and efficiently transport and secretes newly synthesized proteins [42].

TFs are adaptor proteins that recognize and bind to specific DNA sequences, attracting other factors to regulate target genes. This involves local chromatin recruitment of transcriptional coactivators or corepressors, histone modifiers, and nucleosome remodelling proteins [17]. Consequently, functional gene regulatory regions, such as promoters and enhancers, attract large complexes that often assemble in a cell-type-specific manner [52]. Here, we found a core of TFs related to PDLCs, which includes SP1/GC-box, RUNX, AP1 and TEAD. A similar regulation in PDLCs was previously suggested by Tarkkonen et al. [45]. In addition, some TF motifs were found to be exclusive to l-PDLCs in our study. Specifically, the analysis suggested an involvement of SIX (SIX homeobox, described as a player in myogenic and neuronal differentiation [53]) as well as OLIG (oligodendrocyte transcription factor), related to oligodendrocyte differentiation [54]. These TFs were observed at basal levels; therefore, they are relevant at this stage, but their involvement potentially decreases after osteogenic commitment. The potential relevance for *SIX3* expression as predictive marker for impaired osteogenic potential was further confirmed in the second l-PDLC population. Importantly, l-PDLCs do not show enrichment for TF related to an initial commitment to osteogenesis, as observed in h-PDLCs, SP7 and RUNX2. This reinforces the hypothesis of the initial lack of osteogenic commitment in l-PDLCs.

On the other hand, h-PDLCs appeared to be more dependent on SP1 (Sp1 transcription factor), which is associated with osteogenic differentiation of undifferentiated mesenchymal cells, helping the activation of the expression of MAPK1 (mitogen-activated protein kinase 1) through BMP2 (bone morphogenetic protein 2). In dental pulp cells (DPCs), the absence of its expression inhibited osteogenic differentiation, confirming the importance of SP1 in osteogenic commitment [55] The TF AP1 (activator protein 1), composed by several possible dimer subunits such as Fos, Jun and ATF, is involved in biological processes related to regulation/homeostasis of bone tissue [56]. AP1 forms a complex with RUNX2, which is responsible for the regulation of several known osteoblastic genes, such as *BGLAP*, *BMP2*, *IBSP* and *ALPL* [57], and the silencing of AP1 expression causes decreased activities in ALPL [50] and limited osteogenic differentiation.

In an attempt to identify markers to assist with selection of PDLC isolates with good osteogenic potential, we selected a panel of genes for further confirmation by qPCR using a second l-PDLC population. The panel included a marker of pluripotent state (*OCT4*), neuronal lineage (*SIX3*), adipogenic lineage (*PPARG*) and osteogenic lineage (*SP7*). A combination of high expression of *OCT4*, *SIX3* and *PPARG* indeed shows potential in predicting low osteogenic potential, while *SP7* expression does not appear to be a good predictor of high osteogenic potential.

In conclusion, we investigated the epigenetic and transcriptional patterns between PDLCs presenting distinct osteogenic potential. The two populations show different chromatin accessibility and transcriptional maps, with h-PDLCs expressing TFs related to osteogenic commitment; therefore, they are able to facilitate the expression of osteoblastic genes and the deposition of mineral nodules. Conversely, l-PDLCs show hypermethylation of many osteogenic genes, causing a delay in osteoblast differentiation, as demonstrated by the late activation of genes and biological processes already present in h-PDLCs at basal levels. In addition, the unique core of TFs demonstrated in l- and h-PDLCs could contribute in a specific manner to osteoblastic phenotype commitment, and the high expression of pluripotent, neuronal and adipogenic-specific TFs can be predictive of a population with low osteogenic potential.

## Figures and Tables

**Figure 1 cells-11-01126-f001:**
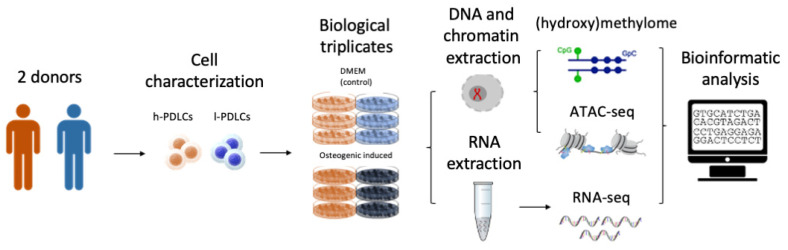
Experimental design. Periodontal ligament cells (PDLCs) were isolated from two different donors and characterized into populations with high osteogenic potential (h-PDLCs) and low osteogenic potential (l-PDLCs) (Assis et al. [5,8]). Three independent experiments were performed for each population where the cells were cultured for 10 days either in standard culture media (DMEM) or in osteogenic media (OM) to promote activation of the pro-osteogenic programme. DNA, chromatin and RNA were collected to perform DNA (hydroxy)methylome (Infinium Methylation EPIC BeadChip (Illumina) assay), Assay for Transposase to Accessible Chromatin (ATAC-seq) and transcriptome (RNA-seq) analysis, respectively.

**Figure 2 cells-11-01126-f002:**
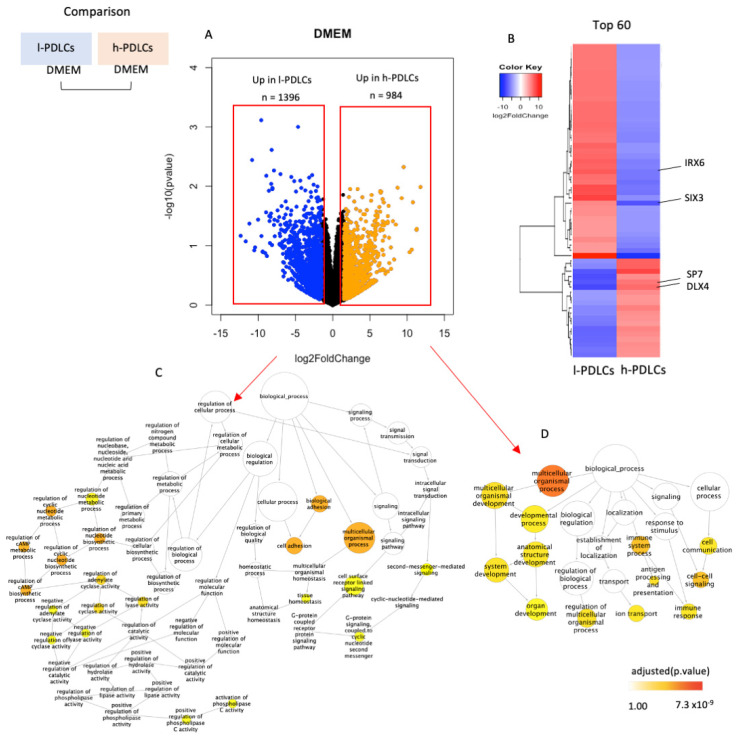
Transcriptional regulation in unstimulated PDLCs with distinct osteogenic potential. RNA-seq data were compared between l- and h-PDLCs cultured in standard DMEM. (**A**) Differential gene expression between the two populations as determined by log2FoldChange. (**B**) The top 60 most differentially expressed genes are shown as a heatmap, highlighting *IRX6*, *SIX3*, *SP7* and *DLX4* genes (red—upregulation; blue—downregulation). (**C**,**D**) Biological processes related to genes upregulated in l-PDLCs (**C**) and in h-PDLCs (**D**) were identified using the Gene Ontology Consortium software followed by the removal of redundant terms using REVIGO. l-PDLCs, periodontal ligament cells showing low osteogenic potential in vitro; h-PDLCs, periodontal ligament cells showing high osteogenic potential in vitro.

**Figure 3 cells-11-01126-f003:**
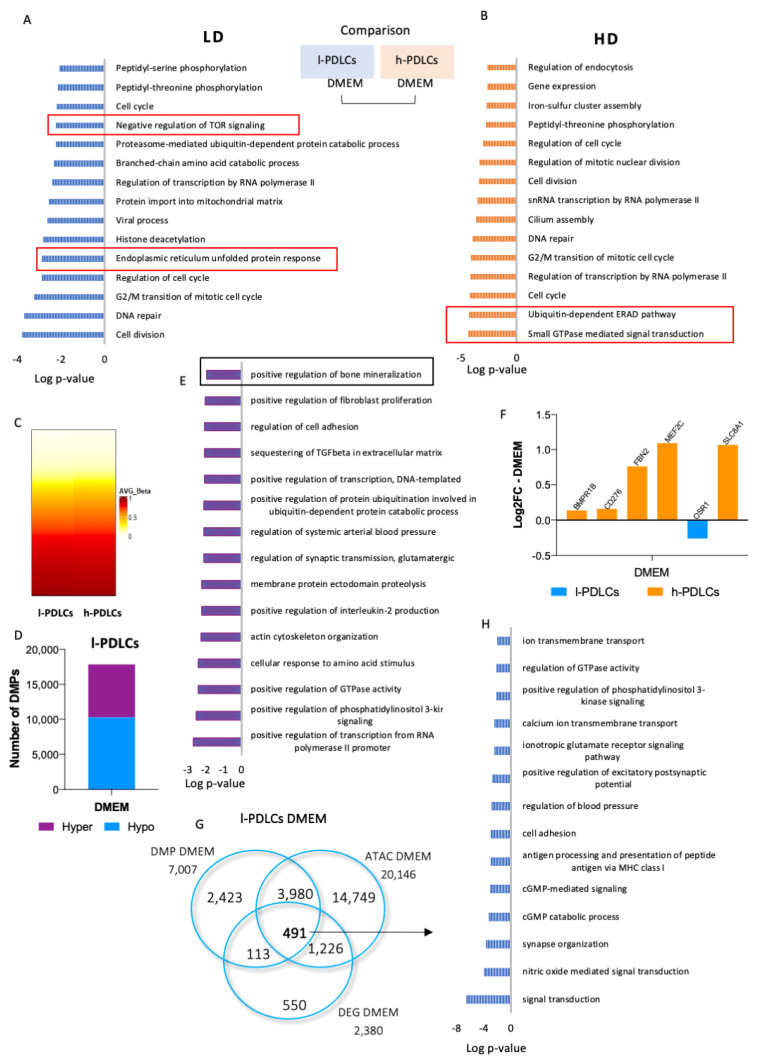
Epigenetic regulation in unstimulated PDLCs with distinct osteogenic potential. l- and h-PDLCs were cultured for 10 days in DMEM for (hydroxy)methylation and chromatin accessibility analyses. (**A**,**B**) ATAC-seq-identified open chromatin regions were linked to a closest gene and gene lists were analysed for pathway enrichment. The analysis was performed using GO Biological Processes in l-PDLCs (LD) (**A**) and h-PDLCs (HD) (**B**). (**C**) DNA methylation was assessed by Infinium Methylation EPIC BeadChip (Illumina). The heatmap shows the resulting methylation Beta values for 850,000 CpG sites in l- and h-PDLCs. The data for hydroxymethylated probes are shown in Appendix A. (**D**) Methylation status (hypomethylation vs. hypermethylation) of 17,847 differentially methylated probes (DMP) in l-PDLCs compared to h-PDLCs. (**E**) Pathway enrichment analysis for genes associated with probes hypermethylated in l-PDLCs. (**F**) Relative transcription levels extracted from the RNA-seq data for genes related to the biological process “positive regulation of bone mineralization”, highlighted in (**E**). (**G**) Overlap of 491 genes identified through upregulated genes, genes linked to accessible chromatin and genes linked to hypomethylated probes in l-PDLCs. (**H**) Top 14 biological process related to overlapped genes in (**G**). Orange: genes upregulated in h-PDLCs; blue: genes upregulated in l-PDLCs. LD, l-PDLCs cultured in DMEM; HD, h-PDLCs cultured in DMEM.

**Figure 4 cells-11-01126-f004:**
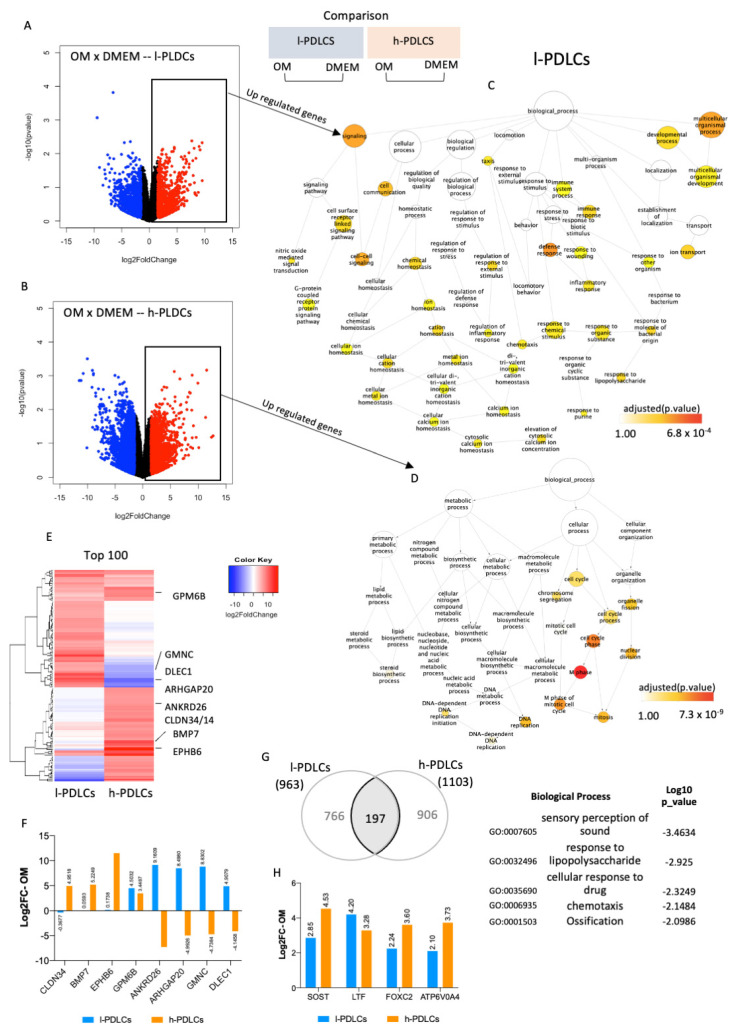
Transcriptional changes in PDLCs with distinct osteogenic potential in response to osteogenic stimulation. Both PDLC populations were cultured in osteogenic media for 10 days after which RNA was extracted for RNA-seq and the results compared to the respective controls (cells cultured in DMEM). (**A**,**B**) Differential gene expression determined by log2FoldChange between basal (DMEM) and stimulated (OM) conditions in l-PDLCs (**A**) and in h-PDLCs (**B**). The genes significantly upregulated upon osteogenic stimulation are shown as red dots. (**C**,**D**) Biological processes related to genes upregulated in l-PDLCs (**C**) and in h-PDLCs (**D**) are shown as network maps. (**E**) Heatmap of the top 100 most OM-upregulated genes in l-PDLCs and h-PDLCs; the genes of interest are indicated to the right. (**F**) Gene expression changes of selected genes after OM stimulation (shown in E) were extracted as Log2FC from the RNA-seq data for both l- and h-PDLCs. (**G**) Overlap of 197 genes commonly affected by OM stimulation in l-PDLCs and h-PDLCs and biological processes most enriched for this gene set (shown to the right). (**H**) OM-induced expression changes of genes related to the biological process “ossification”, highlighted in (**G**), using RNA-seq data (Log2FC). l-PDLCs, periodontal ligament cells with low osteogenic potential in vitro; h-PDLCs, periodontal ligament cells with high osteogenic potential in vitro.

**Figure 5 cells-11-01126-f005:**
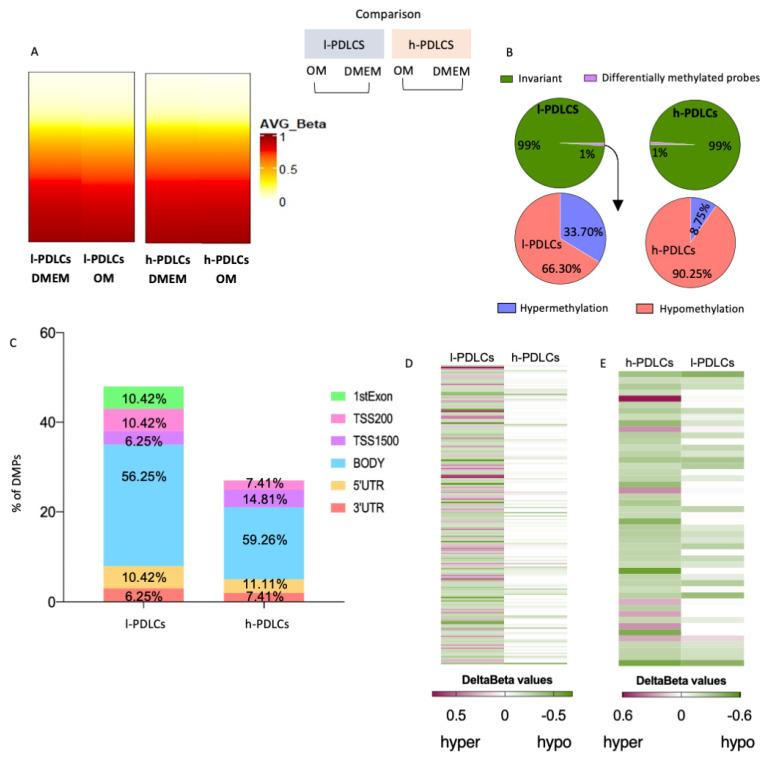
DNA methylation changes upon osteogenic stimulation in PDLCs with distinct osteogenic potential. (**A**) DNA methylation probe data (Infinium Methylation EPIC BeadChip assay) were compared between l- and h- PDLCs cultured for 10 days in OM vs. DMEM (unstimulated controls). Hydroxymethylation data are shown in Appendix A. (**B**) Proportion of differentially methylated probes (DMPs) in l- and h-PDLCs (upper graphs) and distribution of hypo- and hyper-methylated events (lower graphs). (**C**) Genomic localisation of DMPs in l- and h-PDLCs. (**D**,**E**) Heatmaps comparing hyper- or hypomethylated probes in l-PDLCs (**D**) and h-PDLCs (**E**), as determined by delta beta values. Probes differentially methylated in l-PDLCs were identified in h-PDLCs (**D**), and the same was performed for h-PDLCs. (**E**) l-PDLCs/DMEM, periodontal ligament cells with low osteogenic potential at basal levels; h-PDLCs/DMEM, periodontal ligament cells with high osteogenic potential at basal levels; l-PDLCs/OM, periodontal ligament cells with low osteogenic potential at day 10 day of osteogenic stimulation in vitro; h-PDLCs/OM, periodontal ligament cells with high osteogenic potential at day 10 day of osteogenic stimulation in vitro.

**Figure 6 cells-11-01126-f006:**
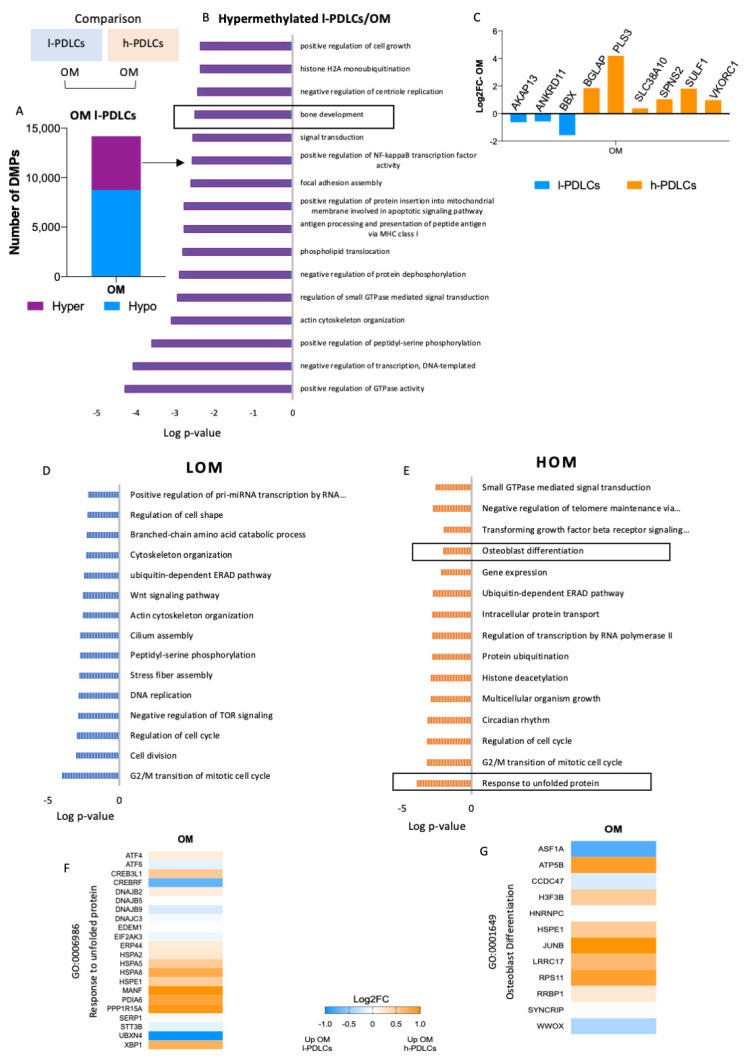
DNA methylation and open chromatin regions in PDLCs with distinct osteogenic potential after osteogenic stimulation. The comparison of l- and h-PDLCs at epigenetic levels (DNA methylation and chromatin accessibility) was performed between data sets obtained after 10 days of cell culture in OM media. (**A**) Proportion of 14,194 differentially methylated probes, either hypo- or hyper-methylated in l-PDLCs as compared to h-PDLCs. (**B**) The top 16 biological processes related to genes hypermethylated in l-PDLCs. (**C**) Gene expression changes related to ‘Bone development’ process highlighted in (**B**), extracted from the RNA-seq data as Log2FoldChange. (**D**,**E**) Open chromatin regions identified by ATAC-seq after osteogenic stimulation were linked to the closest gene, and the gene lists were analysed for pathway enrichment. The analysis was performed using GO Biological Processes and the top 15 terms are shown for l-PDLCs (LOM) (**D**) and h-PDLCs (HOM) (**E**). (**F**,**G**) Differential gene expression levels between l- and h-PDLCs at day 10 of osteogenic stimulation, extracted from the RNA-seq data as Log2FoldChange for genes associated with ‘Response to unfolded protein’ (**F**) and ‘Osteoblast differentiation’ (**G**). Orange: genes upregulated in h-PDLCs; blue: genes upregulated in l-PDLCs. LOM, periodontal ligament cells with low osteogenic potential at day 10 day of osteogenic stimulation in vitro; HOM, periodontal ligament cells with high osteogenic potential at day 10 day of osteogenic stimulation in vitro.

**Figure 7 cells-11-01126-f007:**
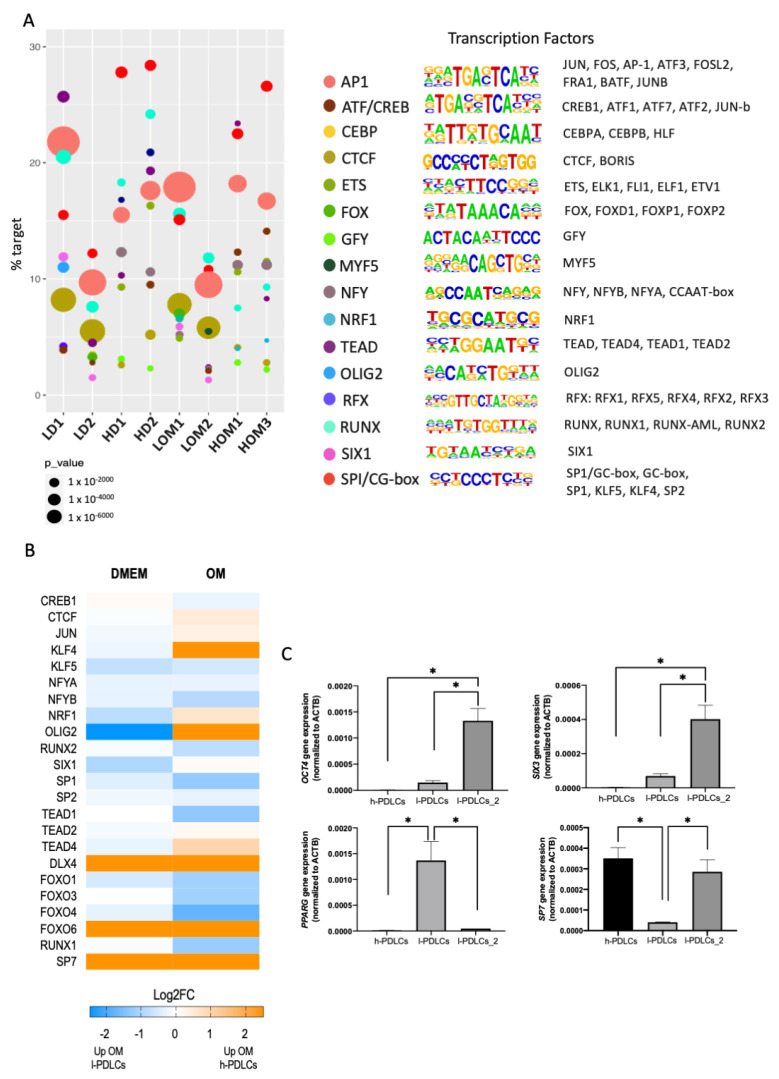
Transcription factor analysis in PDLCs with distinct osteogenic potential. (**A**) ATAC-seq peaks data were used to identify enriched TF motifs in l-PDLCs and h-PDLCs either at basal level or following osteogenic stimulation. The TFs that matched the same de novo motif, were grouped under one identification (as shown to the right), and the results were plotted for each condition and biological repeat (1 or 2). The motifs are sorted according to their frequency detected in ATAC-seq peaks (% target) while the circle size represents the level of significance (*p*-value). (**B**) The expression levels of potential TFs were extracted from RNA-seq data as Log2FC. (**C**) Gene expression levels of *OCT4*, *SIX3*, *PPARG* and *SP7* were analysed in h-PDLCs, l-PDLCs and l-PDLCs_2 to test their predictive osteogenic value at basal levels. Gene expression levels were analysed by qPCR using △△Ct method. N = 3, values are displayed as means–SD. Asterisks above the bars represent significant inter-group differences by ANOVA One Way followed by the Tukey test.

## Data Availability

The data presented in this study are available on request from the corresponding authors. The data are not publicly available due to potentially identifiable genomic information.

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
