# Peer review of "Osteogenic Commitment of Human Periodontal Ligament Cells Is Predetermined by Methylation, Chromatin Accessibility and Expression of Key Transcription Factors"

_cells, 2022, doi:10.3390/cells11071126_

Round 1

Reviewer 1 Report

  1. In DeSeq2 analysis, an adjusted p-value (<0.05) should be used instead of the p-value to reduce false positives. Please provide the number of significant up and down-regulated genes when <0.05 adjusted p-value is applied (with mentioned log2FC).
  2. PDL cells from two donors were pooled and then cultured for all experiments? If not, please describe in detail the procedure.
  3. Please provide PCA plots of seq analysis to show how samples were distinguished.

Author Response

Point 1: In DeSeq2 analysis, an adjusted p-value (<0.05) should be used instead of the p-value to reduce false positives. Please provide the number of significant up and down-regulated genes when <0.05 adjusted p-value is applied (with mentioned log2FC).

Response 1: We agree with the reviewer that an adjusted p-value would allow for a better assessment of the data. However, although RNA was extracted from three biological replicates, these replicates were then pooled for each experimental condition and sequenced as a single sample (see additional explanation in point 2). The lack of sequenced replicates did not allow us to statistically assess the data and instead the genes with log2 fold change above 1.5 or below -1.5 were considered differentially expressed, as described in the Methods section. Pooling the three replicates together should at least partially eliminate the genes with inconsistent, variable expression changes.

Point 2: PDL cells from two donors were pooled and then cultured for all experiments? If not, please describe in detail the procedure.

Response 2: Thank you for giving us the opportunity to clarify this. PDL cells from two donors were not pooled for the experiments. After the cells were collected from the donors, they were induced towards osteogenic differentiation and characterized by flow cytometry, population doubling time, alizarin red and gene expression (Assis et al. 2021. DNMT1 inhibitor restores RUNX2 expression and mineralization in periodontal ligament cells. DNA Cell Biol. 2021;40(5):662-674. doi: 10.1089/dna.2020.6239) (1). Based on this, the two PDLCs populations were classified into high and low osteogenic potential, i.e., the population from one donor showed higher capacity to produce mineral nodules in vitro, when compared to the PDLCs population from the second donor. Next, we performed three independent biological experiments involving 10 days of osteogenic stimulation for each one of the PDLCs (three for h-PDLCs and three for l-PDLCs). The obtained samples were then processed to perform the DNA (hydroxy)methylome, the ATAC-seq and the RNA-seq. In detail, the samples of the three independent biological experiments (3 samples DMEM high-PDLCs, 3 samples DMEM l-PDLCs, 3 samples OM h-PDLCs and 3 samples OM l-PDLCs) were used for the DNA (hydroxy)methylome. For the RNA-seq, the samples of the three independent biological experiments of each experimental condition were pooled, i.e., 3 samples of DMEM h-PDLCs resulted in one sequenced pooled sample for DMEM high. For ATAC-seq analysis, two out of three replicates were chosen for sequencing based on the pre-sequencing controls. We have now clarified our approaches in the Methods section.

Point 3: Please provide PCA plots of seq analysis to show how samples were distinguished

Response 3: Thank you for giving us the opportunity to include this analysis. The data from the four pooled RNA-seq samples (l-PDLCs/DMEM, l-PDLCs/OM, h-PDLCs/DMEM and h-PDLCs OM) were interrogated for principal component analysis (PCA). The plotPCA function from the DESeq2 package was used to perform the PCA and plot the top two principal components. The variability represented by the first component clearly separates the h-PDLCs after OM stimulation from the other samples. This is now included in Supplementary Figure 3 and in the Methods section.

References

  1. Assis RIF, Schmidt AG, Racca F, da Silva RA, Zambuzzi WF, Silvério KG, et al. DNMT1 Inhibitor Restores RUNX2 Expression and Mineralization in Periodontal Ligament Cells. DNA Cell Biol. 2021;00(00):1–13.

Reviewer 2 Report

Introduction. The authors should consider including in this section other studies that show the importance of DNA methylation on modulating osteogenesis in dental stem cells.  I would suggest highlighting the importance of the epigenetic study in PDLSC regarding BMSCs or DPSCs, which exhibit a better mineralization response when compared to PDLSCs.   Results: Can the authors give a little more detail about the obtention of PDLCs with low and higher osteogenic capacity?. Were separated for negative or positive MSCs markers or stemness expression, senescence, or multipotency?  It is known that cells derived from PLSCs or DPSCs contain different stem cells populations, and these can exhibit properties. In this regard, heterogeneous MSCs when fail to differentiate, for instance, into osteoblast, might be a consequence of the potential of stem cell niche within the tissue of origin.   • Lines 255-: Among their biological features, PDLSCs can generate odontoblast or periodontal osteoblast; however, interestingly, l-PLSCs, based on the RNA-seq data or DNA methylation patterns, exhibits a possible neuronal commitment when cultured under maintaining conditions. Could this behavior be a consequence of their ectodermal origin?   • Line 321. The authors claim that poor osteogenic differentiation of l-PDLSC is due to “impairment” of transcriptional activation (see lines 512, 543, 548, 570. Respectfully, and from my point of view, I don´t consider that fact as an “impairment”. In contrast, I believe that there exists or persists an epigenetic memory in those stem cells populations which gives rise to a different commitment, as you commented before on lines 317-318.   Please, consider modifying figures 3F and 6C by bar charts for a better understanding of the transcriptional changes between l- and h-PDLSCs.      Discussion: There is a clear correlation among molecular techniques used here that corroborate some difficulties of l-PDLSCs to differentiate into osteoblast. l-PLSCs appear to require a wide chromatin remodeling and consequence more time before promoting higher expression of osteogenic-selective genes concerning h-PLSCs. With the hypermethylation of osteogenesis-related genes, what is the role of DNMTs and TETs during the induction of PDLSCs toward osteogenesis?  and how both DNMTs and TETs could influence chromatin accessibility?  

The following reports can help to support your findings of the neuronal and adipogenic-specific TFs in l-PLSCs: https://doi.org/10.1038/s41598-019-54745-3; https://doi.org/10.3390/jpm11080738

Minor typos.

  • Doble checks references throughout the manuscript to be consistent with the brackets format. Some e.g can be found in lines 59, 70, 71 among others.
  • Format marks in figures 3, 4, 6, and 7 should be deleted.

Author Response

Point 1: Introduction: The authors should consider including in this section other studies that show the importance of DNA methylation on modulating osteogenesis in dental stem cells. I would suggest highlighting the importance of the epigenetic study in PDLSC regarding BMSCs or DPSCs, which exhibit a better mineralization response when compared to PDLSCs.

 Response 1: Thank you for the suggestion. We added a paragraph showing the importance of DNA methylation on modulating osteogenesis in dental stem cell and epigenetic studies in PDLCSs. In our previous publication Ferreira et al., 2021(Genome-Wide DNA (hydroxy)methylation reveals the individual epigenetic landscape importance on osteogenic phenotype acquisition in periodontal ligament cells; J Periodontology; DOI: 10.1002/JPER.21-0218) (2) we highlighted the role of epigenetic landscape in periodontal ligament cells and the importance of epigenetic regulation (DNA methylation) on osteogenic differentiation. There is currently limited information regarding direct comparison between epigenetic landscapes of DPSCs and PDLSCs, which guided our comparison to BMSCs, known as gold standard for osteogenic differentiation.

Point 2: Results: Can the authors give a little more detail about the obtention of PDLCs with low and higher osteogenic capacity? Were separated for negative or positive MSCs markers or stemness expression, senescence, or multipotency? It is known that cells derived from PDLSCs or DPSCs contain different stem cells populations, and these can exhibit properties. In this regard, heterogeneous MSCs when fail to differentiate, for instance, into osteoblast, might be a consequence of the potential of stem cell niche within the tissue of origin.

 Response 2: We would like to thank the reviewer for pointing this out. Both PDLCs were characterized in our previous publication (3), however the analysis of the expression or lack of expression of specific cell surface markers, such as CD166, CD34 and CD45 were not shown then. The levels of CD34 and CD45 were very similar between h- and l-PDLCs showing less than 1% of expression of positive cells in 10,000 events. Regarding the multipotency marker CD166, both populations also presented similar results with more than 90% of positive cells. However, we agree with the reviewer’s comment that these cells can still display heterogenous populations which we further exemplified by characterizing the populations with low and high osteogenic potential. In this manuscript we aim to extend the understanding of this heterogeneity and identify markers to enable identification of populations with required properties prior to potential applications.

The information regarding the expression or lack of expression of CD166, CD34 and CD45 are now added in the manuscript.

Point 3: Line 255: Among their biological features, PDLSCs can generate odontoblast or periodontal osteoblast; however, interestingly, l-PLSCs, based on the RNA-seq data or DNA methylation patterns, exhibits a possible neuronal commitment when cultured under maintaining conditions. Could this behavior be a consequence of their ectodermal origin?

Response 3: This is a very interesting comment. We found this observation surprising at first but agree with the reviewer that ectodermal origin could provide an explanation.  During embryogenesis, the ectoderm is responsible for formation of neural crest and oral cavity. Since dental cells originate from neural crest, this explains the ability to differentiate into neuronal cells (4). The presence of stem cell niche with neuronal commitment within PDL has also been previously suggested (5). We have now included a comment regarding the neuronal commitment in the Discussion.

Point 4: Line 321. The authors claim that poor osteogenic differentiation of l-PDLCs is due to “impairment” of transcriptional activation (see lines 512,543, 548, 570). Respectfully, and from my point of view, I don´t consider that fact as an “impairment”. In contrast, I believe that there exists or persists an epigenetic memory in those stem cells populations which gives rise to a different commitment, as you commented before on lines 317-318.

 Response 4: We would like to thank the reviewer for this comment. Our research group agreed with this view, where a persistent epigenetic memory might influence differences in commitment between both populations. In our previous publication (Assis et al., 2021: DNMT1 inhibitor restores RUNX2 expression and mineralization in periodontal ligament cells. DNA Cell Biol. 2021;40(5):662-674. doi: 10.1089/dna.2020.6239) (1), we used an epigenetic modulator (DNMT1 inhibitor) to enhance the osteogenic potential in l-PDLCs, confirming the reversibility of epigenetic patterns in genes related to osteogenic differentiation. Therefore, we believe such “impairment” can be reversible using compounds modifying the epigenetic patterns associated with initial cell commitment, especially when genes crucial for the specific differentiation are the ones being modified (i.e. RUNX2 in the early stages of osteogenic differentiation), We have now modified the wording throughout the manuscript.

Point 5: Please, consider modifying figures 3F and 6C by bar charts for a better understanding of the transcriptional changes between l- and h-PDLSCs.

Response 5: Following this suggestion, we created new bar charts to replace the heatmaps in figures 3F and 6C.

Point 6: Discussion: There is a clear correlation among molecular techniques used here that corroborate some difficulties of l-PDLSCs to differentiate into osteoblast. l-PLSCs appear to require a wide chromatin remodeling and consequence more time before promoting higher expression of osteogenic-selective genes concerning h-PLSCs. With the hypermethylation of osteogenesis-related genes, what is the role of DNMTs and TETs during the induction of PDLSCs toward osteogenesis? and how both DNMTs and TETs could influence chromatin accessibility?

Response 6: Thank you very much for giving us the opportunity to clarify this. In our previous publication (1), we investigated gene expression levels of both DNMTs and TETs in the two PDLCs populations.  In general, we observed higher levels of expression of DNMTs and TETs in l-PDLCs than in h-PDLCs, at basal levels (DMEM) and also at early stages of osteogenic induction. These data agree with the hypothesis that l-PDLCs require greater amounts of the epigenetic machinery enzymes, as well as more extensive chromatin remodeling than h-PDLCs. After l-PDLCs were treated with the demethylating agent RG108, which blocks the active sites of DNMT1, a decrease in DNMTs levels, and an increase in TET1 and TET3 levels, were observed. In addition, the DNA methylation levels of the RUNX2 gene were reduced, accompanied by a dominance of this protein in the nucleus, leading to an increase in osteogenic potential of l-PDLCs. In addition, the kinetics (days 3, 10 and 21 of osteogenesis) of several major osteogenic and multipotent genes were investigated and showed the demethylating agent accelerated the osteogenic program. Together, these data suggest an important role of DNMTs and TETs during the induction of PDLCs toward osteogenesis, perhaps accompanied by chromatin remodeling. We have previously shown an association between low methylation levels and chromatin accessibility (6) and are currently investigating, through ChIP-seq, the histone modifications and transcription factors’ binding before and after osteogenic stimulation and as a result of using epigenetic modifiers.  

References

  1. Assis RIF, Schmidt AG, Racca F, da Silva RA, Zambuzzi WF, Silvério KG, et al. DNMT1 Inhibitor Restores RUNX2 Expression and Mineralization in Periodontal Ligament Cells. DNA Cell Biol. 2021;00(00):1–13.
  2. Ferreira RS, Assis RIF, Feltran G da S, do Rosário Palma IC, Françoso BG, Zambuzzi WF, et al. Genome‐wide DNA (hydroxy) methylation reveals the individual epigenetic landscape importance on osteogenic phenotype acquisition in periodontal ligament cells. J Periodontol [Internet]. 2021 Aug 16;55(11). Available from: https://onlinelibrary.wiley.com/doi/10.1002/JPER.21-0218
  3. Assis RIF, Feltran G da S, Silva MES, Palma IC do R, Rovai ES, Miranda TB de, et al. Non-coding RNAs repressive role in post-transcriptional processing of RUNX2 during the acquisition of the osteogenic phenotype of periodontal ligament mesenchymal stem cells. Dev Biol [Internet]. 2021 Feb;470(September 2020):37–48. Available from: https://linkinghub.elsevier.com/retrieve/pii/S0012160620302839
  4. Mayo V, Sawatari Y, Huang C-YC, Garcia-Godoy F. Neural crest-derived dental stem cells—Where we are and where we are going. J Dent [Internet]. 2014 Sep;42(9):1043–51. Available from: http://dx.doi.org/10.1016/j.jdent.2014.04.007
  5. Bueno C, Martínez-Morga M, Martínez S. Non-proliferative neurogenesis in human periodontal ligament stem cells. Sci Rep [Internet]. 2019 Dec 2;9(1):18038. Available from: http://www.nature.com/articles/s41598-019-54745-3
  6. Wiench M, John S, Baek S, Johnson TA, Sung M, Escobar T, et al. DNA methylation status predicts cell type-specific enhancer activity. EMBO J [Internet]. 2011 Jun 24;30(15):3028–39. Available from: http://dx.doi.org/10.1038/emboj.2011.210

Reviewer 3 Report

It is a well-done paper. The authors investigate the epigenetic and transcriptional patterns between PDLCs presenting distinct osteogenic potential. This is part of the general cellular physiology: the gene conditions. It could be nice all these statements are related to the proteins level. I propose the authors to check it with other techniques such as western blot tests in this or other paper.
-    Line 251: “multicellular organismal process”. Clarify these words

Author Response

Point 1: It is a well-done paper. The authors investigate the epigenetic and transcriptional patterns between PDLCs presenting distinct osteogenic potential. This is part of the general cellular physiology: the gene conditions. It could be nice all these statements are related to the proteins level. I propose the authors to check it with other techniques such as western blot tests in this or other paper.

 Response 1: We would like to thank the reviewer for the supportive comment. Indeed, we have some ongoing experiments focused on the chromatin accessibility, histone marks and osteogenic transcription factors, through ChIP-seq analysis. We are also planning to perform western blots to confirm the impact of the epigenetic regulators at protein levels. These will be included in future publications..

Point 2: Line 251: “multicellular organismal process”. Clarify these words

 Response 2: Thank you for giving us the opportunity to explain. “Multicellular organismal process” is a term for a biological process defined as any biological process, occurring at the level of a multicellular organism, identified during ontology analysis. In sequence of this biological process, we observe biological processes related to ossification, shared by both PDLCs populations (Figure 4G).